# Dyslexic Readers Improve without Training When Using a Computer-Guided Reading Strategy

**DOI:** 10.3390/brainsci11050526

**Published:** 2021-04-21

**Authors:** Reinhard Werth

**Affiliations:** Institute for Social Pediatrics and Adolescent Medicine, University of Munich, Haydnstr. 5, D-80336 Munich, Germany; r.werth@lrz.uni-muenchen.de; Tel.: +49-1733550232; Fax: +49-308337940

**Keywords:** dyslexia, reading impairment, reading therapy, eye movements, visual attention

## Abstract

Background: Flawless reading presupposes the ability to simultaneously recognize a sequence of letters, to fixate words at a given location for a given time, to exert eye movements of a given amplitude, and to retrieve phonems rapidly from memory. Poor reading performance may be due to an impairment of at least one of these abilities. Objectives: It was investigated whether reading performance of dyslexic children can be improved by changing the reading strategy without any previous training. Methods: 60 dyslexic German children read a text without and with the help of a computer. A tailored computer program subdivided the text into segments that consisted of no more letters than the children could simultaneously recognize, indicated the location in the segments to which the gaze should be directed, indicated how long the gaze should be directed to each segment, which reading saccades the children should execute, and when the children should pronounce the segments. The computer aided reading was not preceded by any training. Results: It was shown that the rate of reading mistakes dropped immediately by 69.97% if a computer determined the reading process. Computer aided reading reached the highest effect size of Cohen d = 2.649. Conclusions: The results show which abilities are indispensable for reading, that the impairment of at least one of the abilities leads to reading deficiencies that are diagnosed as dyslexia, and that a computer-guided, altered reading strategy immediately reduces the rate of reading mistakes. There was no evidence that dyslexia is due to a lack of eye movement control or reduced visual attention.

## 1. Introduction

Dyslexia is regarded as a specific learning disorder. According to the criteria of the Diagnostic and Statistical Manual of Mental Disorders (DSM5) dyslexia is indicated by (1) inaccurate and effortful word reading, (2) difficulty understanding the meaning of what is read and (4) difficulty with spelling that persisted for at least six months and remains below the skills expected for the chronological age. The DSM5 also requires that the difficulties cannot be explained by intellectual disabilities, poor visual or auditory acuity, psychiatric or neurological disorders, psychological adversity, or inadequate educational instruction [1]. According to these criteria, approximately 5–15% of school children in the USA are dyslexic [2,3,4].

To date, the nature of dyslexia is unclear, and reading therapies are still unspecific, long lasting and of limited success. Therapies to improve the reading performance of dyslexics include therapies based on the discrimination of auditory stimuli [5,6,7,8,9,10], Phonological Awareness which comprises different approaches that are intended to promote reading skills [11], decomposing words into syllables and sounds [12,13,14,15,16,17,18], identifying phonems in words [19,20,21], naming letters, objects, numbers and colors [15,22], and rhyming [23]. Other therapies are based on visual movement discrimination training [24,25,26,27,28], a training to improve eye movement control [29,30,31,32], and syllable segmentation [33,34,35,36]. Therapy studies focused on breaking up words into syllables improved reading performance in children. However, since syllables often contain more letters than poorly reading children can recognize simultaneously longer syllables lead to an increased error rate in some children. The therapeutic effect in these therapy studies was far less than in the studies with words broken up into units containing no more letters the subjects could recognize simultaneously [37,38,39]. Some therapies, like Fast For Word training [7,8,9] did not yield reproducible results. Cohen et al. [40] and Gillam et al. [41] found that Fast For Word training was not better than other traditional therapies. A detailed meta-analysis of studies on the effectiveness of this training program showed absolutely no effect [42,43,44]. The effect of PATH therapy [27,28] was based on few patients and could not be replicated [45]. In addition, the therapy rests on assumptions about the role of magnocells in dyslexia that are still a matter of debate [46,47].

Training procedures that improved reading performance required many months of practice. During this time, many influences on reading performance could not be controlled, and the effect size did not exceed Hedges g = 0.9 [48]. It has already been shown that a reading therapy which has a high training effect of Hedges g between 1.4 and 2 [37,38,39] can be achieved within one single session in which all possible influences on reading performance are controlled. Not only a new reading strategy was practiced during this reading therapy. Subjects also simultaneously practiced expanding their field of attention and focusing their attention on a specific area. It cannot be excluded that the control of eye movements was also improved by computerized steering of eye movements during therapy. This coincides with the assumption that practicing eye-movement control alone may improve reading performance [29,30,31,32]. The present study investigated whether reading capacity can be improved in one single session if (1) only the reading strategy is changed, (2) no training is conducted that may improve the ability to expand the field of attention, or to focus attention, and (3) if no eye movement training is carried out. To explore in what way the reading strategy needs to be changed, it must be investigated in how far a dyslexic child’s reading strategy leads to reading errors. To that end, we must investigate whether reading performance improves when readers do not attempt to recognize more letters of a word or word segment at a time than they can recognize simultaneously. Therefore, it is necessary to investigate how many letters dyslexic children can recognize in a pseudoword, and whether their abilities to recognize pseudowords of a given length improve if fixation times are prolonged. To investigate whether premature saccades and too large saccade amplitudes have an impact on reading performance, reading performance with and without computer guidance of eye movements must be examined. To rule out the possibility that an improvement in reading performance is due to an attention training or an eye movement training, neither a training to improve visual attention performance nor an eye movement training was performed.

A pseudoword experiment was conducted to investigate the influence of the number of letters in a pseudoword, the presentation time, the position of letters in a pseudoword, the masking (crowding) effect [49,50,51,52,53,54,55,56,57] and an insufficient ability to expand the visual field of attention [58,59,60,61,62,63,64,65,66,67]. The results demonstrated in agreement with earlier findings [37,38,39] that impaired reading capacity was neither due to a different masking (crowding) effect in the visual field [49,50,51,52,53,54,55,56,57] or to an insufficient ability to expand the visual field of attention [58,59,60,61,62,63,64,65,66,67]. The result of the present study also supports earlier findings of four independent studies with 296 dyslexic children [37,38,39]. It was demonstrated that the rate of correctly recognized pseudowords depends essentially on the lengths of the pseudowords and on the presentation times.

When reading a text the reader must fixate the right location within a word; s/he must fixate a word or word segment for a given time interval, and the reader must be able to visually process several letters simultaneously. After the end of the fixation time that is needed to recognize the word or word segment, saccades of a given amplitude must be programmed and executed. As visual acuity is only sufficiently high in the fovea and the parafoveal area and drops dramatically up to 10 degrees eccentricity, a word to be read must be projected into the area of highest visual acuity in the middle of the retina. Therefore, a word must be fixated so that its image appears in the fovea and parafoveal region. This must be achieved by eye movements that shift the image of the word to be read onto the region of highest visual acuity. This implies that eye movements of the right amplitude must be executed. As these abilities are necessary conditions for correct reading, the absence or an impairment in one or several of these abilities causes reading problems that are diagnosed as dyslexia.

Computer programs which steer a reader´s eye movements have been developed to explore the influence of appropriate eye movements on reading performance. It has been shown that children improve their reading performance after a 30-min training session when their reading eye movements are adjusted by a computer program [37,38,39]. This was achieved with the help of a reading therapy in which the subjects learned to subdivide the text into segments that consisted of no more letters than they could recognize simultaneously, to fixate these segments at the right location, to execute correct reading saccades, to prolong the time during which the gaze was directed to the segments, and to prolong the time needed to retrieve the sounds that belong to letters or a string of letters from memory. In this way impaired abilities that caused haltingly and faulty reading were compensated and reading performance improved whereas a control group which received no reading therapy showed no improvement [37,38,39].

Therapies that attempt to improve diminished brain functions (which were assumed to cause dyslexia) require long-term practice, have limited effects, and often encounter insurmountable biological limits. The present study investigated whether a therapy is effective that does not attempt to improve diminished brain functions, but compensates these with a new reading strategy tailored to the subjects´ individual reading abilities.

## 2. Experiment 1

### 2.1. Materials and Methods

#### 2.1.1. Patients

Reading performance was investigated in a group of 60 German children (40 boys and 20 girls) aged 8 to 15 years (mean age: 122.4 months, SD = 19.3 months) who were diagnosed as dyslexic by the Zuercher Reading Test [68]. All children were native German speakers and attended Bavarian primary schools. Forty children were below the 6th percentile (1.5 SD), and 20 children were below the 2.5 percentile (i.e., 2 SD). All children had a pediatric, an ophthalmological, and a psychological examination. They were right-handed, had no neurological, psychiatric, visual, or auditory deficits and no speech disorders. The children were second-to-tenth graders who knew all individual letters and were expected to read fluently but were far behind the required reading ability. Their reading disabilities were not based on lack of teaching or inadequate educational instructions, as the children had had the same educational instructions as other children in the same grade. The children had been referred to the pediatric clinic (Kinderzentrum München) of the Institute for Social Pediatrics and Adolescent Medicine of the University of Munich because of their reading problems.

The children’s IQ was in the normal range on the Hamburg-Wechsler Intelligence Test for children [69]. All children participated in experiment 1.

#### 2.1.2. Methods

Experiment 1 tested under which conditions poor readers were able to correctly read at least 95% of a list of 20 pseudowords using the Celeco Software-Package for the Diagnosis and Therapy of Dyslexia [70]. For this purpose the length of the pseudowords, the presentation time, and the time to pronounce the pseudowords were altered until all subjects were able to read at least 95% of the pdeudowords correctly [39].

As each letter of the pseudowords corresponded to a different phonem in the German language, it was possible to identify the letters in the pseudowords that had been read incorrectly. A word was considered read incorrectly if at least one letter had been omitted or replaced by a letter not present in the pseudoword, if the location of at least one letter had been changed, or if at least one letter had been incorrectly added.

Lists of 20 pronounceable 2-, 3-, 4-, 5- or 6-letter pseudowords were presented at eye level on a monitor. The sequences of the letters in the pseudowords were also found in colloquial German words. Each of these pseudowords contained the same number of consonants and vowels at the same location within the word.

The distance between the eyes and the monitor was 40 cm. The words were black (luminance of 4 cd/m^2^; altitude 14 mm; space between types: 4 mm) on a background of 68 cd/m^2^. The presentation times of the pseudowords varied between 250 and 500 milliseconds. Luminance and presentation time of stimuli were assessed using a Gigahertz Optimeter P 9201 with a sampling rate of 20 microseconds. Each trial began with the presentation of a green fixation mark (luminance: 30 cd/m^2^; background luminance: 68 cd/m^2^) in the center of the monitor. The child was asked to direct his/her gaze to the fixation mark. When the child maintained fixation, the fixation mark disappeared and was replaced by a pseudoword that was centered at the fixation point. Fixation of the word segments and saccadic eye movements were recorded using an infrared eye-tracking system (IRIS eye tracker; sampling rate: 500 Hz). In the first trial, a sequence of 20 pseudowords consisting of 4 letters was presented. Each pseudoword appeared for 250 ms. The child was instructed to read each pseudoword aloud. If the child was unable to pronounce the word correctly, s/he was asked to spell and write the word. The time between the onset of the presentation of the pseudoword and the onset of the child’s speech was measured by the computer.

If more than one out of 20 words were not read correctly, it was investigated whether a prolongation of the fixation time alone was sufficient to improve the child´s ability to recognize the pseudowords. If 90% (i.e., 18 out of 20 pseudowords) or less of a sequence of pseudowords were read correctly, a different sequence of 20 pseudowords of the same length was presented. The presentation time of each pseudoword in the new sequence was increased by 50 ms. If still less than 90% or less of this sequence of letters was read correctly, a new sequence of 4-letter pseudowords was presented. Each new pseudoword was presented for 350 ms. If 90% or less of a sequence of pseudowords was read correctly, within a presentation time of 500 ms, a different list of pseudowords was presented and the number of letters was reduced by one. Therefore, the fixation times increased and/or the number of letters to be read was increased or decreased until 95% of a list of pseudowords was read correctly.

If more than one of these 20 pseudowords with a length of n letters were not read correctly at a presentation time of 500 ms, the experiment was repeated with a different list of 20 pseudowords with a length of n − 1 letters. Again the presentation times were increased by steps of 50 ms until the child was able to read at least 19 of the 20 pseudowords correctly. If the subject was able to read at least 19 of the 20 pseudowords with a length of n letters that were presented for 250 milliseconds correctly, the experiment was repeated with pseudowords with a length of n + 1 letters. The children were instructed not to start pronouncing until they were sure of the word and not to start pronouncing immediately. To avoid too early pronunciation a sound signal was given 700 ms after the pseudoword appeared. The subjects were not supposed to start speaking until they heard the sound signal. After each pronunciation, the subjects were asked to correct themselves, if necessary, within 5 to 10 s. After an interval of between 5 and 10 s, the green fixation mark was presented again. When the child´s gaze was on the fixation mark, a different pseudoword appeared for the same time interval as the previously shown pseudoword. The children´s reading performance was registered by recording their voice with a microphone. Speech onset, the presented pseudoword, the presentation time of the pseudoword, and the voice of the subject were recorded by a computer. The experiment took no longer than 45 min.

Statistics: Rates of reading mistakes were compared using the Bonferroni-Holm corrected Wilcoxon-test.

### 2.2. Results

The results of experiment 1 are summarized in Figure 1 and Table 1. Dyslexic children differed considerably with regard to the number of letters they were able to recognize and the fixation time needed to recognize a sequence of letters. From the result of experiment 1 it follows that a sufficiently long fixation time is a prerequisite for the children’s ability to recognize pseudowords of a given length. It also follows that reading mistakes occur if pseudowords are too long and readers try to recognize more letters of pseudowords simultaneously than they are able to. Furthermore, reading mistakes may come about when a reader pronounces a word or word segment before the sequence of sounds has been retrieved from memory (too short a speech-onset latency.

For 3- and 4- letter pseudowords, the means of the rate of misread letters increased from positions 1 to 3. In the case of 5-letter pseudowords, the means of the rate of misread letters increased from positions 1 to 5. For 6- letter pseudowords, the means of the rate of misread letters increased from positions 1 to 4 and decreased again from positions 4 to 6 (Figure 1). For 3- and for 6-letter pseudowords, the Bonferroni-Holm corrected *p*-value of the Wilcoxon-test showed that the *p*-value of the difference between the means of the frequencies of misread letters at the 1st, 2nd, 3rd, 4th, 5th, and 6th position within a pseudoword was *p* > 0.1. For 4-letter pseudowords, the difference between the means of the rate of reading mistakes at positions 1 and 3 within a pseudoword was *p* < 0.0001 (Bonferroni-Holm corrected Wilcoxon-test). The difference of the means between the frequency of reading mistakes at the 1st and 4th positions was *p* < 0.0003 (Bonferroni-Holm corrected Wilcoxon-test). The difference of the means of the frequency of reading mistakes at positions 2 and 3 within a pseudoword was *p* < 0.014 (Bonferroni-Holm corrected Wilcoxon-test). For all other comparisons, the *p*-value of the Bonferroni-Holm corrected Wilcoxon-test was *p* > 0.09. For 5-letter pseudowords, the difference between the means of the frequency of reading mistakes at positions 1 and 3, 1 and 4 and 1 and 5 within pseudowords was *p* < 0.0001 (Bonferroni-Holm corrected Wilcoxon-test). Comparison of the means of the rate of misread letters at positions 2 and 4 yielded a Bonferroni-corrected *p*-value of the Wilcoxon-test of *p* < 0.0034. The difference of the means between the frequency of reading mistakes at the 2nd and 4th position was *p* < 0.021 (Bonferroni-Holm corrected Wilcoxon-test). For the comparison of the means of the rates of misread pseudowords at all other positions the *p*-value of the Bonferroni-Holm corrected Wilcoxon-test was *p* > 0.1. Table 1 shows also that the length of the pseudowords had no effect on the speech onset latency: If the speech onset latencies for 3-letter pseudowords, 4-letter pseudowords, 5-letter pseudowords, and 6-letter pseudowords are compared, the Cohen-d effect size shows always no effect (speech onset latencies for 3-letter pseudowords vs. 4-letter pseudowords: Cohen-d = −0.086; CI = −0.059–0.232; CC = 95%; 4-letter pseudowords vs. 5-letter pseudowords: Cohen-d = −0.096; CI = −0.052–0.244; CC = 95%; Cohen-d = −0.086; 5-letter pseudowords vs. 6-letter pseudowords: Cohen-d = 0.151; CI = 0.076–0.378; CC = 95%).

## 3. Experiment 2: Immediate Improvement in Reading Ability after Changing the Reading Strategy

Experiment 2 investigated whether a child’s ability to read a text improves if the computer guides the children’s reading strategy such that (1) the child only attempts to simultaneously recognize words or word segments consisting of no more letters than it is able to recognize simultaneously, (2) the amplitude of the reading saccades does not exceed the number of letters the child is able to recognize simultaneously, (3) the child fixates the word segments for the time interval needed (adequate fixation intervals), and (4) if the time interval between the onset of the presentation of the word and the onset of the pronunciation of the word by the child is sufficiently long (adequate speech-onset latency). The role of an increased fixation time and pseudoword length in improving reading performance had already been investigated in Experiment 1, in which the stimuli were stationary and did not require eye movements. Experiment 2 examined the role of appropriate eye movements on improvement in reading performance. The length of the sequence of letters that could be recognized simultaneously and the required fixation times found in Experiment 1 were transferred to Experiment 2. A colored cursor then indicated the length of word segments and fixation times. Experiment 2 investigated whether reading performance improved when the length of the word segments to be read and the fixation times were adapted to the performance recorded in Experiment 1 and eye movements were guided by the computer.

### 3.1. Materials and Methods

#### 3.1.1. Children with Dyslexia

All children with dyslexia who had participated in Experiment 1 participated in Experiment 2.

#### 3.1.2. Procedure

Half of the children in the therapy group read one half of a text without the help of a computer. The other half of the children in the therapy group read the other half of the texts with the help of a computer. The computer controlled how many letters subjects attempted to recognize simultaneously, the fixation times, the eye movements, and the speech onset latency. The children of the control group read both halves of the text without the support of a computer.

The children were assigned to the therapy group (30 children) or to the control group (30 children) according to their ability to read the letters of pseudowords simultaneously. After each pseudoword experiment, the number of letters a child could recognize simultaneously was known. Children who could recognize the same number of letters simultaneously were assigned to the therapy group or the control group in such a way that there was approximately the same number of children in each group. If several children had the same ability to recognize a certain number of letters simultaneously, the children were assigned to the therapy group and control group in such a way that there were approximately the same number of children in each group who needed the same fixation time. Children who had almost the same ability to recognize a certain number of letters simultaneously and needed the same fixation time were assigned to the therapy group or the control group in such a way that in both groups there were approximately the same number of children who had almost the same age. Thus, the therapy group and the control group were similar in the ability to read letters simultaneously, and in the fixation time they needed to read a given number of letters simultaneously. Table 1 shows the distribution of the children in the therapy group and the control group (mean age in the therapy group: 120.83 months; SD 16.24 months; mean age in the control group: 124.03 months SD: 21.45 months). Comparison of the ages of both groups using the Wilcoxon-test resulted in a *p*-value of 0.5.

The children of the therapy group and of the control group red the same texts. In the therapy session the children of the therapy group read with the help of a computer. Half of the children in the therapy group read the first part of four different texts before the therapy session without the help of a computer and the second part during the therapy session with the help of a computer. The other half of the children in the therapy group read the second part of the texts before the therapy session without the help of a computer and the first half during the therapy session with the help of a computer. Half of the children in the control group read the first part of the texts first and the second part later. The other half of the controls read the second part of the texts first and the first part later. The controls received no support by the computer while reading the same texts as the therapy group. The first part of text 1 consisted of 29 words (142 letters), the second part of this text consisted of 30 words (139 letters). The first part of text 2 consisted of 33 words (156 letters) the second part of this text consisted of 34 words (161 letters). The first part of text 3 consisted of 26 words (129 letters), and the second part of this text consisted of 25 words (130 letters). The first part of text 4 consisted of 20 words (131 letters) the second part of this text consisted of 19 words (127 letters). The experiment took no longer than 30 min.

The children were sitting in front of a monitor. The distance between the eyes and the monitor was 40 cm. The words were black (luminance of 4 cd/m^2^; altitude 14 mm; space between types: 4 mm) on a background of 68 cd/m^2^. Luminance of stimuli and background was measured with a Gigahertz Optimeter P 9201. Fixation of the word segments and saccadic eye movements were recorded using an infrared eye-tracking system (IRIS eye tracker; sampling rate: 500 Hz). Only the therapy group was instructed to apply an adequate reading strategy using the Celeco Software-Package for the Diagnosis and Therapy of Dyslexia [39,70]. To adopt such an adequate reading strategy, children’s reading strategy was guided by a computer program that instructed the reader (1) to read only words or word segments not containing more letters than the children were able to recognize simultaneously according to the result of Experiment 1, (2) to fixate these words or word segments for the appropriate time interval, (3) to start to pronounce the words or word segments only after an appropriate time interval, and (4) to execute eye movements of an amplitude that matches the length of the words or word segments whose letters can be recognized simultaneously (adequate reading saccades).

To support this reading strategy, a yellow mark indicated the point to be fixated within each word or word segment. A green cursor (segment cursor) to the left and/or right of the yellow fixation mark indicated which letters in the word segment were to be read simultaneously together with the letter indicated by the yellow fixation mark. The yellow and green marks indicated which adjacent letters in a word or word segment should be read while fixating the yellow mark. The subjects were to read the text aloud so that reading errors could be recognized immediately by the therapist. Whenever a word segment was recognized, the next word segment was shown. Then the yellow fixation mark was moved to the middle letter of the next word or word segment, indicating the goal of the saccade (i.e., the location where the gaze should be directed when the next word segment has to be read). A green cursor (segment cursor) to the left and/or right of the yellow fixation mark again showed which letters of the newly shown word or word segment were to be read while the eyes fixated the shifted yellow mark. The fixation mark and the segment cursor moved from one word segment to another as they were to be read in succession. The whole text was presented on the monitor. The text to the left of a word or word segment that had to be read was not shown on the monitor to prevent the child from exerting a saccade to the left and refixate a word or word segment that had already been read. Only the text that had not yet been read appeared on the monitor. An acoustic signal was presented 1 s after the yellow and the green cursors had moved to the new segment to be read. The acoustic signal indicated when the subject was allowed to pronounce the word segment to prevent a premature pronunciation of the word segment to be read.

#### 3.1.3. Statistics

Rates of reading mistakes and reading time were compared using the Wilcoxon-test.

To show that the improvement of pseudoword recognition is an effect of the change of the reading strategy, Cohen d effect size statistics [71,72] was used: Cohen $d=\frac{X_{1}-{\;X}_{2}}{\mathrm{Sw}}$
$\mathrm{Sw}=\sqrt{\frac{\left(n_{1}-1\right)S_{1}^{2}+\left(n_{2}-1\right){\;S}_{2}^{2}}{\left(n_{1}+n_{2}-2\right)}}$
$X_{1}\;$ and $X_{2}$ are the means and $S_{1}$ and $S_{2}$ are the standard deviations of the rate of reading mistakes. $n_{1}$ and $n_{2}$ are the number of values from which each mean values was calculated.

### 3.2. Results

The children who read the first or the second half of the text without the help of the computer read in the mean 16.57; SD = 6.76 (which corresponds to 7.67%; SD = 3.13%) of the words incorrectly. When these children read the remaining half of the text with the help of the computer, in the mean only 5.03 words; SD = 3.56 (corresponding to 2.33%; SD = 1.65%) of the words were read incorrectly. This corresponds to a decrease of 11.72 words read incorrectly, i.e., a 69.97% decrease of reading mistakes. The difference between the number of reading mistakes for non-computer supported reading was *p* ≤ 0.00001 according to the Wilcoxon Test. The effect size Cohen d was 2.137 (CI = 1.24–3.034; CC = 95%).

In the control group the rate of reading mistakes increased when the subjects read one half of the text first and when reading the remaining half afterwards. When reading one half of this test first, a mean of 14.4 words (6.67%); (SD = 4.0 (1.85%)) were read incorrectly. When the controls read the remaining half of this test later on, 17.2 (7.96%) of the words (SD = 3.43%) were read incorrectly (*p* > 0.1: Wilcoxon Test). The effect size Cohen d was 0.512 (CI =−0.215–1.239; CC = 95%). Thus, a comparison between the performance of the two halves of the children in the control group showed a weak effect. A comparison between the therapy group and the control group (Cohen d _therapy group_–Cohen d _control group_) showed an effect size of 2.649. This is the highest effect size that has ever been measured for a reading therapy. This high effect size was already reached immediately after changing the reading strategy without any reading training.

Children who read half of the text without the help of the computer took 158.32 (SD = 61.98) seconds to read the texts. When the children read with the help of the computer, it took them 342.43 (SD = 77.89) seconds. Comparison of both time intervals with the Wilcoxon-test resulted in *p* < 0.00001. When the control group read one half of the text first they needed 138.57 (SD = 64.15) seconds. When they read the remaining text they needed 164.2 (SD = 84.74) seconds. Comparison of both time intervals with the Wilcoxon-test showed a *p*-value of *p* < 0.00001. Whereas there was no difference between the therapy group and the control group when they read the first part of the text without the help of a computer (Wilcoxon-test: *p* > 0.05), there was a marked difference between the reading times of the control group when reading the second half of the texts and the reading times of the therapy group who read the second half of the texts guided by the computer (Wilcoxon-test: *p* < 0.00001).

## 4. Discussion

The aim of Experiment 1 was to investigate how many letters pseudowords can contain to be recognized simultaneously and how long each individual child must fixate pseudowords of a given length to recognize them. The children were assigned to the therapy group or to the control group according to their abilities to simultaneously recognize a string of letters in pseudowords and the fixation times that they needed. This ensured comparability of the therapy group and the control group. The result of Experiment1 determined the positions and lengths of the yellow and green cursor in Experiment 2. The cursor indicated how many letters the children should try to read at a time to simultaneously recognize a string of letters. As the amplitudes of the saccades that must be executed to connect each word or word-segment to be read without a gap between them depend on the number of letters each child can recognize simultaneously, the result of Experiment 1 is also presupposed to guide the childrens’ reading eye movements by the computer.

The results of experiment 1 demonstrate that dyslexic readers can read at least 95% of pseudowords correctly if they fixate them at the right location, the length of the words is adjusted to the subjects ability to simultaneously recognize letters, and the fixation time and the speech-onset latency are prolonged. This is not due to lack of knowledge of the grapheme-phoneme correspondence. All subjects who participated in experiment 1 were familiar with the grapheme-phoneme correspondence of all letters and the knowledge of this correspondence was unimpaired in all subjects. No practice was necessary to improve the subjects’ abilities to read pseudowords. Experiment 1 was performed with pseudowords because they cannot be guessed, and each letter must be recognized. When reading normal words, they can be guessed if only a few letters are recognized. Table 1 shows that the ability to recognize a succession of letters simultaneously, the length of the fixation times needed to recognize a given string of letters, and the length of the speech-onset times that the subjects needed to pronounce at least 95% of the pseudowords correctly differed considerably among subjects. Whereas five subjects were able to recognize 6 letters simultaneously (Table 1), 19 subjects were only able to recognize 3 letters simultaneously, 18 subjects were able to recognize 4 letters simultaneously, and 18 subjects were to recognize 5 letters simultaneously. If a subject who is only able to recognize 3 letters simultaneously tries to recognize 6 letters simultaneously s/he will make reading mistakes: S/he will swap letters, displace letters, omit letters, and read letters that do not occur in the word. The finding that recognition depends on the presentation time of stimuli is in agreement with psychophysical [73,74,75,76,77,78,79] and neurobiological studies [80] and earlier studies on the recognition of pseudowords [37,38,39]. Temporal summation explains both the improvement in recognizing a sequence of letters with increased fixation time [37,38,39] and poor readers’ prolonged fixation times reported in previous studies [81,82,83,84,85]. Poor readers tend to misread words more often than good readers. If they misread a word, they often notice that the misread word is not a meaningful word or that the misread word does not fit into the context of the sentence in which it occurs. To correct this mistake they do what everyone does when s/he assumes that s/he has not recognized something correctly: S/he directs his/her gaze longer at the object to be recognized and focuses his/her attention on it. However, this is not sufficient to correctly recognize the word if the fixation time is still too short or if s/he tries to recognize more letters simultaneously than s/he can.

In Experiment 1 the presentation time was limited to 500 ms, because temporal summation is effective up to this fixation time [73]. The results of Experiment 1 and earlier studies on temporal summation [37,38,39,73,74,75,76,77,78,79] support the hypothesis that improvement of word recognition requires prolongation instead of shortening [32,86,87,88] of the fixation time. The finding that readers improve when they extend their fixation times demonstrates that attention does not decrease during fixation, and that the readers can maintain their attention for the required fixation time. This contradicts the assumption that poor reading is caused by an attention deficit. If readers had been unable to maintain attention, the rate of misread letters would have increased as the fixation time had increased.

As reading mistakes occurred not only at the beginning and at the end of words, but at all positions within words, a reduced ability to recognize a succession of letters simultaneously cannot only be attributed to a narrowing of the field of attention [58,59,60,61,62,63,64,65,66,67]. The area in which humans can detect and recognize visual stimuli varies depending on where attention is focused. As early as 1909, Balint [89] demonstrated that subjects could narrow or widen their field of attention depending on the extension of the object they were watching. In 1917, Poppelreuter [90] showed that patients with a normally extended visual field may still be unable to recognize objects next to the object on which they are focusing their attention. The objects further from the fixation point were only recognized with longer fixation times. Poppelreuter described this as a narrowing and widening of the field of attention and called this phenomenon „a disturbance of overview“. Williams and Gassel [91] also showed that the visual field narrows if a subject directs his/her attention vigorously to a point in the middle of the perimeter used to assess the extension of the visual field. Subsequent studies have confirmed these results and have shown that the retinal area in which many stimuli can be detected simultaneously narrows when subjects focus their attention on a given point in the visual field [58,59,60,61,62,63,64,65,66,67]. It has also been shown that the field of attention can be shifted to all areas of the visual field regardless of eye movements [92,93,94]. In agreement with earlier studies [37,38,39] the present study demonstrates that dyslexic children have different abilities to simultaneously recognize a given number of letters in pseudowords. The ability to recognize a given number of letters improves when presentation time is prolonged. If a narrowing of the field of attention was the cause of reading mistakes, most errors would be expected to occur at the beginning and the end of the pseudowords. Figure 1 demonstrates that this was not the case. The results of experiment 1 also show that poor reading performance is not due to a different masking (crowding) effect [49,50,51,52,53,54,55,56,57,58,59]. If this was the case one would assume that letters in the middle of pseudowords which are masked by other letters to the left and to the right (crowding effect) are more often misread than letters at the right end of the pseudowords. However, subjects misread letters at the end of pseudowords which are not masked by other letters on both sides, more often than letters in the middle of pseudowords which are masked on both sides irrespective of the word length (Figure 1). The result also shows that letters at the fixation point and immediately to the right of the fixation point were not misread more frequently than letters at the fixation point or further to the right of the fixation point. This does not support the assumption that dyslexics suffer from an unusual foveal or parafoveal processing [48,49,50,51,52,53,54,55,56]. Children made less reading errors for letters at the beginning than at the end of the words. These results are in agreement with the result of earlier studies [37,38,39]. Table 1 also shows that the number of letters that can be recognized at the same time depends on the fixation time. If the fixation time is prolonged more letters can be recognized simultaneously. This finding contradicts the assumption that attention was not focused on the area in the visual field where the pseudowords were presented: Each trial began with the presentation of the fixation point, and it was controlled whether the subjects directed their gaze steadily to the fixation point. The children knew where the pseudoword would appear, and they could focus their attention on this area before the pseudoword was presented. The subjects were also able to read 95% of the pseudowords correctly without visual attention training and without improving their visual attention capabilities. Therefore, reduced ability to simultaneously recognize all letters in a pseudoword should not be regarded as a consequence of an impaired visual attention span [61,62,65,66,95,96,97,98]. Many poor readers improved their ability to simultaneously recognize a string of letters by applying a longer fixation time, i.e., an increased temporal summation [73,74,75,76,77,78,79]. Nineteen children were unable to recognize pseudowords consisting of 4 or 5 letters even if the fixation time was prolonged up to 500 ms. These children have a more severe impairment of simultaneous recognition that cannot be compensated by an increased temporal summation. A reduced simultaneous-recognition capacity is different from an attention disorder and should be regarded separately.

The results of experiment 2 demonstrate that a reduction in the word length, a prolongation of the fixation time, and computer guidance of reading eye movements are sufficient to drastically improve reading performance. The rate of reading errors immediately dropped by 69.97%, and the effect size of the computer-guided reading strategy was Cohen d = 2.649. The subjects in the control group who read without computer assistance showed no improvement in reading performance.

### 4.1. The Role of Eye Movements in Dyslexia

The question of whether and to what extent irregular eye movements contribute to a reading disorder is still a matter of debate [29,30,31,32,39,81,82,83,84,85,99,100,101,102,103,104,105,106]. The importance of eye movements for reading results from the distribution of visual acuity in the visual field. Visual acuity is highest in the fovea and a small paravoveal area and drops dramatically towards the periphery. At 5 degrees of eccentricity, visual acuity drops to 50%, and at 10 degrees eccentricity, there is only about 30% visual acuity left [107]. Therefore, we are only able to read the text of a book if words we want to read are projected onto the area of sufficiently high visual acuity in succession. If the words a person reads exceed the area of sufficiently high acuity, letters outside that area cannot be recognized.

An ideal reader executes reading saccades that match the number of letters s/he can recognize simultaneously in the reading direction. If the amplitude of reading saccades exceeds the number of letters that a person can recognize simultaneously, reading mistakes are inevitable. If a reader who recognizes only 3 letters simultaneously exerts a saccade over 7 letters, s/he can only recognize three letters before performing the saccade, and s/he can read three letters after having completed the saccade. Then there is a gap of four unrecognized letters between the three letters that were read before and the three letters that were read after the saccade.

An ideal reader executes a succession of staircase-like saccades in the reading direction. Saccades opposite to the reading direction (regressions) may occur in normal readers and are frequent in dyslexics [39,81,82,101]. A reader may exert staircase like reading saccades the amplitude of which do not exceed the number of letters that the person is able to recognize simultaneously. This person’s reading performance may not be hindered by regressions if they are interspersed in correct staircase like reading saccades. Regressions may occur in the presence of suitable eye movements in the reading direction without causing poor reading performance. It may be more difficult for the eye to find the correct target of a reading saccade if one or more regressions occur before the saccade to the correct location in the word to be read next is completed. It has been shown that reading therapy is successful even if regressions are interspersed in correct reading saccades after reading therapy [39]. In the present study the text that had already been read was deleted such that regressions were prevented. Guiding eye movements and preventing regressions increased reading performance more than when regressions were present as was the case in an earlier study [39]. The present study demonstrates that inappropriate eye-movements impair reading performance if saccades in the reading direction are initiated too early, and if their amplitude exceeds the number of letters that the reader can recognize simultaneously [38,39]. If patients are unable to direct their gaze on a word or word segment for at least 200 ms or to execute eye movements of an appropriate amplitude, reading errors will occur. All children in the present study and in previous studies in which they learned an appropriate eye movement strategy [37,38,39] were able to fixate words for a sufficiently long time interval and perform appropriate eye movements. This means that the children did not have a reduced ability to fixate words or to perform appropriate saccades [81,102,103] but that they used an incorrect eye movement strategy.

The result of experiment 2 shows that the reading performance of dyslexic subjects improves immediately without any training if their reading strategy is guided by a computer. The computer shows them (1) where a word segment to be read should be fixated, (2) how many letters s/he should try to recognize simultaneously, (3) how long s/he should fixate each segment, (4) that the amplitude his/her reading saccades should not exceed the number of letters that the subject can recognize simultaneously and where the goal of each reading saccade should be, (5) that the subject should not exert eye movements opposite to the reading direction, and (6) how long the time between the beginning of the fixation of a word segment and its pronunciation should be. Such a reading strategy must take into account the individual’s abilities to simultaneously recognize a given number of letters, the fixation time and speech-onset time needed to recognize and pronounce a given word segment. The results show that under these reading conditions the rate of reading mistakes decreased dramatically and that such a computer guided reading yields the highest effect size that has ever been measured in a reading therapy.

In languages such as German, Italien and Spanish there is a close grapheme-phoneme correspondence. There is no reason to assume that reading strategies differ significantly in languages with about the same grapheme-phoneme correspondence. Eye movement records of normal readers whose mother tongue is English show the same staircase-like eye movements as normal German readers [39,82,84,99,108,109]. English-speaking readers, like German readers, must shift the word segments to be read with saccades directed to the right in the foveal and parafoveal area. The sequence of saccades in reading direction and fixation phases is a necessary and sufficient condition for reading, presupposing the reader has a normally developed visual system. Eye movements that are neither necessary nor sufficient for improving reading performance are inappropriate or superfluous. Only when readers split the text into small word segments, the pronunciation of which depends on the letters that follow the word segment to be pronounced, as it is often the case in English, other eye movements are required. In this case, the reader must first look to the word segment to be pronounced, then to the following word segment, and finally back to the word segment to be pronounced.

### 4.2. Does Slow Reading Improve Reading Performance?

Slow reading may be due to long fixation times or to searching eye movements during reading (Figure 2a: A and B). The results of experiment 1 show that reading performance improves if readers prolong the time interval which they fixate the word or word segment to be read. But even if fixation time is prolonged, the number of letters that a reader can recognize simultaneously differs between persons.

Slow reading may also be due to searching eye movements in and against the reading direction that are executed during a long time interval before the subjects start to pronounce the text to be read (see Figure 2). These readers’ eye movements to the right typically exceed the number of letters that can be recognized simultaneously, and fixation times are shorter than they need to recognize a given succession of letters. Therefore, slow reading improves reading performance only if the reader does not try to recognize more letters than s/he can and if s/he does not scan the text for a long time with inadequate searching eye movements. In the present study, computer aided reading was slower than free reading because the children were instructed not to pronounce the words or word segments to be read before they were sure that they had recognized the word or word segment correctly (speech onset latency). Even if they were prone to pronounce a word or word segment after 600 or 800 ms, they were not allowed to do so because they were only allowed to pronounce the word or word segment after a tone signal had been presented 1 s after the beginning of the fixation period. After the children had heard the tone signal they could only start to pronounce the word or word segment after a given reaction time of at least 250 ms. Prolongation of the speech onset latency was important because many children are liable to make reading mistakes when they start to pronounce before the sequence of sounds has been retrieved correctly from memory. Prolongation of speech onset latency prevents premature pronunciation.

## 5. Conclusions and Outlook for Future Research

The results of the present study demonstrate that reading problems are not due to a lack of eye movement control or reduced visual attention. The results suggest that poor reading is caused by an inappropriate eye movement strategy which consists of executing a saccade before the word or word segment to be read has been recognized. This means that the fixation times required to recognize the words or word segments are too short. Reading mistakes occur if the saccades executed to the next word or word segment to be read are greater than the number of letters that can be recognized simultaneously. As a result, recognized words or word segments do not connect without gaps, and letters in the gaps are overlooked. When reading aloud, readers begin pronunciation too early, i.e., before the sound sequence associated with the word to be read has been correctly retrieved from memory. When these causes of reading errors are eliminated by a new, computer-guided reading strategy, the rate of misread words immediately drops dramatically.

The computer program used in the present study has also been successfully used to teach dyslexics a reading strategy tailored to their individual abilities and needs. The children reduced their rates of reading errors by nearly two-thirds in one session [37,38,39]. To further improve reading performance, parents were instructed to conduct reading training with this computer program daily. Parents reported that the children continued to improve their reading skills. If the children are no longer under the therapist’s control in everyday life, it is not possible to determine which factors other than the reading training influence their reading performance. It would be desirable to conduct the study over a longer period of time, during which all influences on reading performance can be controlled.

## Figures and Tables

**Figure 1 brainsci-11-00526-f001:**
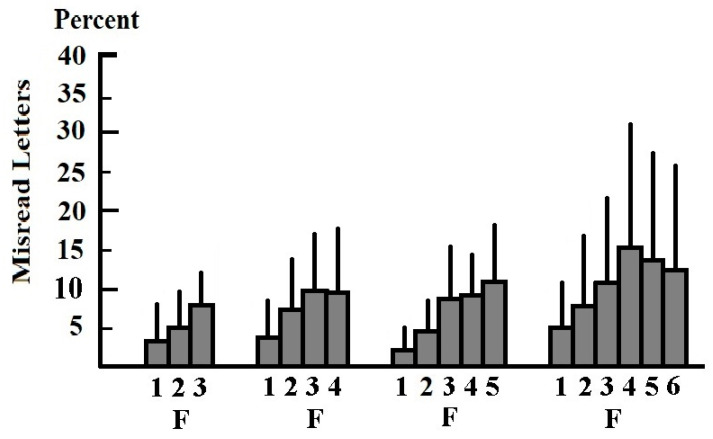
Percentage of incorrectly read letters in 3-letter, 4-letter, 5-letter, and 6-letter pseudowords according to their location in the words. Vertical bars above columns indicate standard deviation. The letters were either omitted, replaced by other letters, or shifted to an incorrect position. Numbers denote the letters in the pseudowords from left to right. F denotes the letter that occurred at the fixation point. The rate of incorrectly read letters increased from left to right. The rate of reading mistakes was lowest for the first letter. In 3-letter and 4-letter pseudowords it was highest for the third letter, in 5-letter pseudowords for the fifth letter, and in 6-letter pseudowords for the fourth letter.

**Figure 2 brainsci-11-00526-f002:**
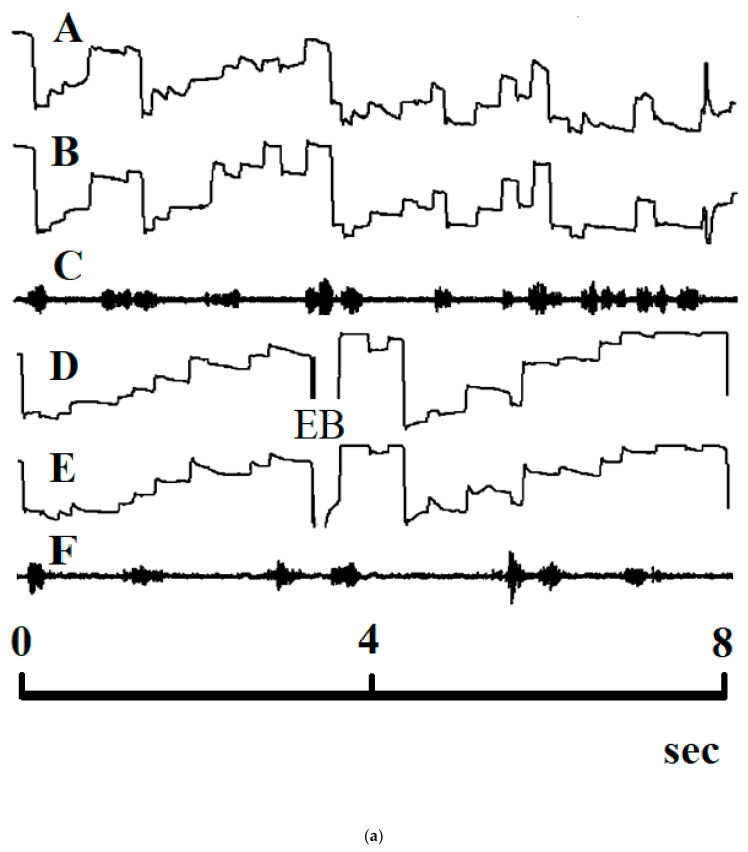
(**a**,**b**): Searching eye movements of a subject who participated in experiment 2 with many regressions during free reading. D and E: The subject exerts staircase-like eye movements with only one regression and longer fixation times during computer-guided reading. C: Speach spectogram during free reading; F: Speach spectogram during computer-guided reading. The speech spectogram (F) shows that the subject pronounces more slowly during computer-guided reading than during free reading (C). Figure 2 B: A and B: eye movements of a dyslexic reader who participated in experiment 2 during free reading. The subject exerts mainly staircase-like eye movements. D and E: staircase-like eye movements of the same subject during computer-guided reading with only one regression (arrow) and longer fixation times. C: Speach spectogram during free reading; F: Speach spectogram during computer guided reading. The speech spectrogram (F) shows that the subject pronounces more slowly during computer-guided reading than during free reading (C). Even if fixation time is prolonged up to 500 ms, some readers are unable to recognize more than 3 or 4 letters simultaneously. If slow reading is due to long fixation times, reading performance improves only if the reader does not try to recognize more letters simultaneously as s/he is able to. Otherwise long fixation times do not improve reading performance.

**Table 1 brainsci-11-00526-t001:** The number of letters (columns 2–5, from left to right), fixation times (first column on the left) and mean speech onset times (bottom row) at which 60 dyslexic children were able to read at least 95% of the pseudowords correctly. First column on the left: presentation times (i. e. fixation times) of the pseudowords; second column: number of subjects who were able to read 3-letter pseudowords within fixation times between 250 and 500 ms; third column: number of subjects who were able to read 4-letter pseudowords within fixation times between 250 and 500 ms; fourth column: number of subjects who were able to read 5-letter pseudowords within fixation times between 250 and 500 ms. Fifth column: number of subjects who were able to read 6-letter pseudowords within fixation times between 250 and 500 ms. Bottom row: means and standard deviations of speech onset latencies. TG and CG indicate the number of children who belonged to the therapy group (TG) or control group: (CG).

Fixation Time Milliseconds	Number of Letters Recognized			
	3 Letters	4 Letters	5 Letters	6 Letters
	Number of Subjects Who Recognized > 95% of the Pseudowords Correctly			
250 ms	TG: 3 CG: 3	TG: 2 CG: 3	TG: 3 CG: 4	TG: 2 CG: 1
300 ms	TG: 2 CG: 3	TG: 1 CG: 1	TG: 3 CG: 2	
350 ms	TG: 1 CG: 2	TG: 3 CG: 2	TG: 1 CG: 1	
400 ms	TG: 1 CG: 1	TG: 1 CG: 1	TG: 1 CG:	TG: 1 CG: 1
450 ms	TG: 1 CG:	TG: CG: 1	TG: CG:	
500 ms	TG: 1 CG: 1	TG: 2 CG: 1	TG: 1 CG: 2	
∑ Subjects	TG: 9 CG: 10	TG: 9 CG: 9	TG: 9 CG: 9	TG: 3 CG:2
Speech Onset Latency	X = 1456.45 ms SD = 473.08 ms	X = 1404.84 ms SD = 705.60 ms	X = 1466.39 ms SD = 562.96 ms	X = 1393.86 ms SD = 484.52 ms

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
