# Peer review of "Dyslexic Readers Improve without Training When Using a Computer-Guided Reading Strategy"

_brainsci, 2021, doi:10.3390/brainsci11050526_

Round 1
Reviewer 1 Report
Although research into improvement of reading of Dyslexic readers is imported. This study need to be improved seriously.
The introduction starts with unclear and fragmented criteria for dyslexia. I recommend authors either to use the full-cited text or translate the criteria to a full sentence.
Furthermore, the theoretical underpinnings for this research are thin. A clear research question is missing in the introduction. It’s stays unclear why two experiments has been set up, what the aim of these experiments are and which hypotheses are tested.
In this section 2.1.1 the participants, the methods and materials merge into each other. Make a clear distinction between these aspects. With respect to the participants, there is a huge difference in age and graders. On what theoretical ground is chosen for such a great difference in age and graders. In the last sentence of the first paragraph of section 2.1.1. Authors claim that ‘the reading disabilities were not based on lack of teaching or inadequate educational instructions’. On what ground can they claim this? How are the children recruited, how many schools were involved?
The method is described in great detail, however a theoretical underpinning for is method is lacking.
The statistics section lack body and content.
Authors claim that the results suggest that poor reading is caused by an inappropriate eye movement strategy and a reduced ability to simultaneously recognize a sequence of letters. This is a huge claim based on a research without strong theoretical underpinnings and executed on a very small and a diverse research group of dyslectic children and a control group of non-dyslectic children is lacking.
Author Response
Response to reviewer 1
Reviewer 1: The introduction starts with unclear and fragmented criteria for dyslexia. I recommend authors either to use the full-cited text or translate the criteria to a full sentence.
Response: The text has been reworded.
Reviewer 1: Furthermore, the theoretical underpinnings for this research are thin.
Response: The history of science shows that important scientific questions may be asked and answered without theoretical underpinnings. The question „why does the apple fall from the tree“ or „what is the nature of light“ can be asked without any knowledge of a physical theory. Nevertheless, the resulting research may lead to the development of Newtonian mechanics or quantum mechanics. Furthermore, the referee´s criticism is vague because it is not clear what the referee understands by „theoretical underpinnings“. If one specifies the concepts „theory“ or „theoretical“, the criticism proves to be unjustified.
One type of scientific theories may consist of assertions about the applicability of mathematical structures, such as Einstein's field equations on objects in space and time or Schroedinger's wave equation on photons and electrons. After the applicability of these theories had been proven many times, relativity theory became relativity mechanics and quantum theory became quantum mechanics. Then, these theories were no longer theoretical.
Statements that result from experimental findings but have not yet been tested experimentally, may also be termed „theories“. In dyslexia research, these are statements such as "the distribution of visual acuity in the retina, temporal summation, simultaneous recognition, the field of attention, eye movements etc., may have a fundamental role in reading." The objectives and methods employed in the present study are based on a number of experimental results as evinced by more than 100 references. The references also show that the methods used in the present study rest on findings about the distribution of visual acuity in the visual field and its physiological and psychophysical basis, eye movements, experimental results about the field of attention, simultaneous recognition, temporal summation of visual stimuli, and on the successful use of these methods in earlier studies. The questions investigated in the present work are therefore very well founded by experimental findings and „theories“ in the latter sense of the word.
Reviewer 1: A clear research question is missing in the introduction. It’s stays unclear why two experiments has been set up, what the aim of these experiments are and which hypotheses are tested.
Response: The introduction has been revised.
Reviewer 1: In this section 2.1.1 the participants, the methods and materials merge into each other. Make a clear distinction between these aspects.
Response: A new caption has been added.
Reviewer 1: With respect to the participants, there is a huge difference in age and graders. On what theoretical ground is chosen for such a great difference in age and graders. In the last sentence of the first paragraph of section 2.1.1. Authors claim that ‘the reading disabilities were not based on lack of teaching or inadequate educational instructions’. On what ground can they claim this? How are the children recruited, how many schools were involved?
Response: There is no reason to limit the children who participated to a given grade. If only children in a given grade participated one could argue that the results only apply to children in this grade. From a methodological perspective, it is essential that the children in the therapy group correspond to the children in the control group. This can only be achieved if the groups are paralleled i. e. if children assigned to the control group are selected according to their abilities tested in the pseudoword test. Assignment of children to the therapy group and the control group is described in detail in the methods section of Experiment 2 (page 14). I have added text that highlights this aspect on page 18.
As described on page 6, the children were recruited according to their performance in the Zuerich Reading Test if their reading abilities were 1.5 or 2 standard deviations below the performance of children in the same grades.
All children attended Bavarian primary schools and received the same reading instructions as the good readers.
Reviewer 1: The method is described in great detail, however a theoretical underpinning for is method is lacking.
Response: The criticism is unfounded. The methods employed in the present study are based on a number of experimental studies as evinced by more than 100 references. The references also show that the methods used in the present study rest on research about the distribution of visual acuity in the visual field, eye movements, the field of attention, simultaneous recognition, temporal summation of visual stimuli, and on the successful use of these methods in earlier studies.
Reviewer 1: The statistics section lack body and content.
Response: What does it mean that the statistic section lacks body and content. The Wilcoxon test and the Bonferroni-Holm corrections are well known and widly employed. Their use does not require justification. The use of Cohen effect size statistics is the statistical method of choice to demonstrate the therapy effect and meets the requirements of the American Statistical Association. I have added the mathematics of this statistics on page 17.
Reviewer 1: Authors claim that the results suggest that poor reading is caused by an inappropriate eye movement strategy and a reduced ability to simultaneously recognize a sequence of letters. This is a huge claim based on a research without strong theoretical underpinnings and executed on a very small and a diverse research group of dyslectic children and a control group of non-dyslectic children is lacking.
Response: That poor reading is caused by an inappropriate eye movement strategy and a reduced ability to simultaneously recognize a sequence of letters is definitely not „a huge claim based on a research without strong theoretical underpinnings…“ This claim rests on the fact that visual acuity is only sufficiently high in the fovea and a small paravoveal area and drops dramatically towards the periphery. Therefore, eye movements must shift the words to be read in the center of the visual field where acuity is sufficient. I have added a text that highlights this aspect (pp. 22-23).
The literature concerning the field of attention and simultaneous recognition cited in the text also shows that the area in which we can recognize letters is limited by the expansion of the field of attention. The underpinning of this finding rests on numerous experimental results, many of which are cited in the text. I have added text that highlights this line of research (pp.20-21). Balint (1909) already showed that subjects can narrow or widen their field of attention depending on the dimensions of the object they are watching. Poppelreuter (1917) has shown that patients with a normally extended visual field may be unable to recognize objects next to the object on which attention is focussed. Objects further off the fixation point were only recognized when fixation times were longer. Williams and Gassel (1962) showed that the visual field narrows if a subject directs his/her attention vigorously to a point in the middle of the perimeter used to assess the extension of the visual field. Many investigations (e. g. Chaikin 1962; Engel 1971; Ikeda and Takeuchi 1975) and numerous more recent articles have shown that the area of the retina in which many stimuli can be detected simultaneously narrows when the subjects must focus their attention in the middle oft he visual field. The area where items are detected widens if the subjects don´t need to focus their attention on a given point of the retina, and if they can can spread their attention in a wider area. These studies show that the number of items which can be detected or recognized simultaneously depend on how much attention must be focussed on the center of the visual field. Many investigations (e.g. Chaikin 1962; Engel 1971; Ikeda and Takeuchi 1975) and numerous more recent studies have shown that the retinal area in which many stimuli can be detected simultaneously narrows when the subjects must focus their attention on the middle of the visual field. Posner (1980) and Jonides 1981) showed that the field of attention can be shifted in any area of the visual field without exerting eye movements. We have shown in earlier studies (Werth 2006, 2018.2019; Klische 2007; and the present study) that dyslexics have different abilities to recognize a given number of letters in pseudowords. Our studies studies confirm that the ability to recognize a given number of letters improves when the presentation time is prolonged.
Regarding the repeated criticism of a lack of theoretical underpinnings, what was said above about theories applies. This criticism is baseless.
The criticism that the claim is based „… on a very small and diverse research group of dyslexic children…“ does not apply. The claim rests on the result of 5 independent studies (including the present one) in which 356 dyslexic children participated. Compared to other studies on dyslexia this is not a small, but one of the largest groups of children which has been investigated. The result was also repeatable in 5 independent studies. The studies are well founded and meet the requirements of the American Statistical Association according to which studies must have repeatable results. In an earlier eye movement study I have demonstrated that reading performance in children with dyslexia improves (effect size Hedges g = 1.4) more than in other therapies if the amplitudes of saccades are adjusted to the children´s abilities to simultaneously recognize a string of letters and the children execute more saccades of smaller amplitudes (Werth 2019).
The criticism that „… a control group of non-dyslectic children is lacking“ is incorrect from a methodological point of view. To test a therapy effect, dyslectics with and without therapy must be compared, not dyslectics with good readers. A study with good readers is an inappropriate control study for what was investigated in the present study. The questions adressed in the present study are (1) how many letters dyslexic children can recognize in a pseudoword, (2) whether their abilities to recognize pseudowords of a given length improve if fixation times are prolonged, and (3) which role do eye movements play in reading. The methods appropriate for answering these questions were described in the text. To investigate pseudowords of which length good readers can recognize does not answer questions adressed in the present paper.
Reviewer 2 Report
This is a very interesting paper that aims at testing, based on an individual approach, the impact of a reading-aid program for children with dyslexia. Results show that children using the computer-based reading aid program committed significantly fewer reading errors than children reading without help with a large effect size. One of the most important aspect is that the program helps children with dyslexia to solve their specific reading difficulties as determined in Experiment 1. However, in my view, this important aspect is not sufficiently emphasized in the paper. The importance of the data collected in Experiment 1 for adapting the computer-aid reading program to the specific need of each child could be highlighted. Moreover, it would be interesting to know whether these effects are long-lasting? In other words, did children read better without aid after reading with the computer aided-reading program or did they go back to their "normal" reading level? Even if this was not directly addressed in this study, this should be discussed or at least mentioned in the perspectives for future work.
Introduction
The introduction provides relevant information and is clealry written.
p.2, end of 3rd paragraph: As clearly stated in the introduction, several abilities are necessary for correct reading and a deficit in one, or several, of these abilities is possibly associated with reading problems. The problem is how to specify which ability (or abilities) is deficient? How to determine which ability needs to be compensated?
p.2 last paragraph and top of p.3: Results and conclusions are presented at the end of the introduction. In my view, it would be more appropriate to detail the hypothesis at the end of the introduction and to keep this paragraph for the conclusion at the end of the discussion.
Methods
Overall, more information is needed on the diagnosis of dyslexia: how many children had phonological deficits? visual attention deficits? co-morbidity such as attention deficits, etc...
Experiment 1
Table 1 is difficult to understand. Within the table it is indicated “Number of Subjects who Recognized > 95% of the Pseudowords Correctly” and in the legend one can read “second column: number of subjects who were able to read 3-letter pseudowords within fixation times between 250 and 500 ms” but then “third column: number of subjects who were unable to read more than 4-letter pseudowords within fixation times between 250 and 500 ms; fourth column: number of subjects who were unable to read more than 5-letter pseudowords within fixation times between 250 and 500 ms. Fifth column: number of subjects who were unable to read more than 6-letter pseudowords within fixation times between 250 and 500 ms.” Pls clarify and please comment these results in the text: the number of children with dyslexia who were able (or unable?) to correctly read the pseudo-words is very low in both groups…
Figure 1: Results for 6-letter pseudo-words showed a decrease for positions 5 and 6: pls comment.
Experiment 2
Because all these different factors are manipulated at the same time, it is difficult to determine which manipulation is most effective: pls comment.
Pls replace “patients” by “children with dyslexia”.
p.7, first paragraph: there are redundancies in this paragraph that need to be suppressed.
2nd paragraph: again, to avoid too much redundancy, it may be easier to write something like “the texts that were read with and without the help of the computer were balanced across children of the therapy group.’ The same remark holds for children in the control group.
End of 4th paragraph: how did the computer program “forced” the children to adopt a particular strategy? Pls explain in more detail.
p.8, 1st paragraph: “Only the text that had not yet been read appeared on the monitor.” Was the text to be read presented all together or one word at a time on the computer screen?
p.8, Results: text reading is much longer with than without the help of the computer: is it time only that helps the reader or what children must do based on the computer-aid program?
p.9, Discussion: “All subjects who participated in experiment 1 were familiar with the grapheme-phoneme correspondence of all letters and the knowledge this correspondence was unimpaired in all subjects. “ Was the grapheme-phoneme correspondence checked for all children and for all combinations present in the PW?
“Poor readers often fixate a word to be read longer than good readers do because they are aware that they need a longer fixation time.” How are they aware? The logic seems unclear here.
“As word recognition improves with prolongation of fixation time, improvement of word recognition requires prolongation instead of shortening [32, 77-79] of the fixation time”: again, it seems that there is a problem in the writing here. May be “Results support the hypothesis that improvement of word recognition requires prolongation instead of shortening [32, 77-79] of the fixation time.”
“The least letters were misread at the beginning of the word regardless of word length.”: could be simpler to write “children made less reading errors for letters at the beginning than at the end of the words.” Except for 6-letters words: pls comment.
“Therefore, reduced ability to simultaneously recognize all letters in a pseudoword should not be regarded as a consequence of an impaired visual attention span [61, 62, 65, 66, 80-83]. Many poor readers improved their ability to simultaneously recognize a string of letters by applying a longer fixation time, i.e., an increased temporal summation [73-76].” This argument is unclear because a longer fixation time may induce more efficient visual attention processes: pls comment.
“Some children were unable to recognize pseudowords consisting of 4 or 5 letters even if the fixation time was prolonged up to 500 ms. Pls specify how many children?
p.10, bottom: “Such a reading strategy must take into account the individual’s abilities to simultaneously recognize a given number of letters, the fixation time and speech-onset time needed to recognize and pronounce a given word segment. “ As mentioned in the general comment above, this is a very good point.
Figure 2: it seems that there are some mistakes in the legend: Figure 2A: “The speech spectrogram (E) shows that the subject pronounces more slowly during computer-guided reading.” The speech spectrograms seem to be C and F (rather than E) with F slower than C. Similar comment for Figure 2B: “
As mentioned in the general comment a general conclusion is needed with a summary of the main findings, their interpretation and the perspectives for future work.
Author Response
Response to reviewer 2
I should like to thank reviewer 2 for substantial and helpful suggestions.
Reviewer 2: This is a very interesting paper that aims at testing, based on an individual approach, the impact of a reading-aid program for children with dyslexia. Results show that children using the computer-based reading aid program committed significantly fewer reading errors than children reading without help with a large effect size. One of the most important aspect is that the program helps children with dyslexia to solve their specific reading difficulties as determined in Experiment 1. However, in my view, this important aspect is not sufficiently emphasized in the paper. The importance of the data collected in Experiment 1 for adapting the computer-aid reading program to the specific need of each child could be highlighted.
Response: I have added a text at the beginning of the discussion in which I highlighted the importance of the results of Experiement 1.
Reviewer 2: Moreover, it would be interesting to know whether these effects are long-lasting? In other words, did children read better without aid after reading with the computer aided-reading program or did they go back to their "normal" reading level? Even if this was not directly addressed in this study, this should be discussed or at least mentioned in the perspectives for future work.
Response: A text dealing with this issue has been added at the end of the discussion (pp. 29-30).
Reviewer 2: The introduction provides relevant information and is clealry written.
p.2, end of 3rd paragraph: As clearly stated in the introduction, several abilities are necessary for correct reading and a deficit in one, or several, of these abilities is possibly associated with reading problems. The problem is how to specify which ability (or abilities) is deficient? How to determine which ability needs to be compensated?
Response: The question of whether an ability is defective often means whether this ability is below a given norm. This was, however, not the approach in the present experiments. If pseudowords of a certain length cannot be recognized at a certain fixation time, but are recognized at a longer fixation time, the reader must fixate for a correspondingly longer time. If s/he cannot recognize the pseudowords even at 500 ms fixation time, the potential of temporal summation is exhausted. This means that the reader cannot recognize pseudowords of a certain length even with a long fixation time. Therefore, the pseudowords must be shortened by at least one letter. Fixation times and/or eye movements are defective if a reader does not fixate a word or word segment as long as s/he should and if s/he tries to read more letters simultaneously than s/he can. The question is not whether these abilities correspond to a norm, but at which fixation times and word lengths this individual child is able to recognize 95 % of a series of 20 pseudowords. The reader must perform eye movements that lead to correct fixation and that match the length of pseudowords s/he can recognize simultaneously. If the eye movemnents do not match his/her abilities, they are defective. The role of increased fixation times and pseudoword lengths in improving reading performance was investigated in Experiment 1. In Experiment 1 the stimuli are stationary and did not require eye movements. Experiment 2 examined the role of appropriate eye movements in improving reading performance. The length of the sequence of letters that can be simultaneously recognized and the fixation times required found in Experiment 1 were transferred to Experiment 2 in which a colored cursor indicated the length of word segments and fixation times. Experiment 2 investigated whether reading performance improved when the length of the word segments to be read and the fixation times were adapted to the performance found in Experiment 1 and when eye movements were guided by the computer.
Reviewer 2: p.2 last paragraph and top of p.3: Results and conclusions are presented at the end of the introduction. In my view, it would be more appropriate to detail the hypothesis at the end of the introduction and to keep this paragraph for the conclusion at the end of the discussion.
Response: This paragraph was moved to the end of the discussion as requested (pp. 29-30).
Methods
Reviewer 2: Overall, more information is needed on the diagnosis of dyslexia: how many children had phonological deficits? visual attention deficits? co-morbidity such as attention deficits, etc...
Response: As mentioned in the "Patients" section, none of the children had phonological deficits, visual attention deficits, a hyperactivity disorder, etc... I added that all children had had a pediatric, an ophthalmological, and a psychological examination.
Experiment 1
Reviewer 2: Table 1 is difficult to understand. Within the table it is indicated “Number of Subjects who Recognized > 95% of the Pseudowords Correctly” and in the legend one can read “second column: number of subjects who were able to read 3-letter pseudowords within fixation times between 250 and 500 ms” but then “third column: number of subjects who were unable to read more than 4-letter pseudowords within fixation times between 250 and 500 ms; fourth column: number of subjects who were unable to read more than 5-letter pseudowords within fixation times between 250 and 500 ms. Fifth column: number of subjects who were unable to read more than 6-letter pseudowords within fixation times between 250 and 500 ms.” Pls clarify and please comment these results in the text: the number of children with dyslexia who were able (or unable?) to correctly read the pseudo-words is very low in both groups…
Response: The mistakes in the description of Table 1 have been corrected (pp. 10-11).
Only children who were 1.5 or 2 SD below normal readers in the same grade participated in the study. Most poor readers were unable to recognize many letters simultaneously, i. e. could not read even 4- or 5-letter pseudowords. Only few children were able to read 95% of 6-letter pseudowords correctly.
Reviewer 2: Figure 1: Results for 6-letter pseudo-words showed a decrease for positions 5 and 6: pls comment.
Response: For 6-letter pseudowords, the difference between the rate of misread letters at positions 5 and 6 was always p>0.1. I assume that this is just a statistical effect because readers who were tested with 6-letter pseudowords were already able to read 5-letter pseudowords and did not misread many 6-letter pseudowords. Here I followed the recommendations oft he American Statistical Association and did not interpret p-values. Such interpretations in terms “significant” or “non-significant” have been criticized in modern statistics.
Experiment 2
Reviewer 2: Because all these different factors are manipulated at the same time, it is difficult to determine which manipulation is most effective: pls comment.
Response: The role of increased fixation times and pseudoword lengths in improving reading performance was already investigated in Experiment 1where the stimuli were stationary and did not require eye movements. Experiment 2 examined the role of appropriate eye movements on improving reading performance. The length of the sequence of letters that could be simultaneously recognized and the fixation times required found in Experiment 1 were transferred to Experiment 2 in which a colored cursor indicated the length of word segments and fixation times. Experiment 2 investigated whether reading performance improved when the length of the word segments to be read and the fixation times were adapted to the performance found in Experiment 1 and when eye movements were also guided by the computer. This Text has been added on page 13.
Reviewer 2: Pls replace “patients” by “children with dyslexia”.
Response: Has been replaced as requested (p. 13).
Reviewer 2: p.7, first paragraph: there are redundancies in this paragraph that need to be suppressed.
Response: From my point of view, the experimental setup is somewhat complex. The referee read the text very carefully but not all readers do so. To avoid misunderstandings, it is advisable to repeat some aspects. I would, therefore, prefer not to shorten the text.
Reviewer 2: 2nd paragraph: again, to avoid too much redundancy, it may be easier to write something like “the texts that were read with and without the help of the computer were balanced across children of the therapy group.’ The same remark holds for children in the control group.
Response: I am not sure if the readers kept in mind the way the texts were balanced. To avoid misunderstandings, I would, therefore, prefer not to shorten this text. I deleted redundant text on p. 23.
Reviewer 2: End of 4th paragraph: how did the computer program “forced” the children to adopt a particular strategy? Pls explain in more detail.
Response: "Forced" is indeed a bit too strong a term. The children, of course, could not be forced. I have therefore replaced "forced" with "instructed" (p. 8).
Reviewer 2: p.8, 1st paragraph: “Only the text that had not yet been read appeared on the monitor.” Was the text to be read presented all together or one word at a time on the computer screen?
Response: The whole text was presented. Only the text that had already been read was deleted to prevent saccades opposite to the reading direction. This information was added on p. 16.
Reviewer 2: p.8, Results: text reading is much longer with than without the help of the computer: is it time only that helps the reader or what children must do based on the computer-aid program?
Response: I explained this in more detail in additional text that was added in section „Does slow reading improve reading performance?”, pp. 27-28.
I extended the text on p. 29: Text reading takes longer because the readers are instructed to fixate the words or word segments to be read longer than they fixate the words or word-segments when reading without the help of the computer. The children had been instructed not to start to pronounce until they are sure that they had recognized the word or word segment (speech onset latency). To avoid too early pronunciation, the children were only allowed to pronounce the phonems corresponding to the string of letters after they heared the sound that indicated that they were allowed to pronounce the word or word-segment. The sound appeared 1 second after the beginning of the fixation of a word or word segment. Longer fixation times and longer speech onset latencies slowed reading, but reduced the rate of misread words.
Reviewer 2: p.9, Discussion: “All subjects who participated in experiment 1 were familiar with the grapheme-phoneme correspondence of all letters and the knowledge this correspondence was unimpaired in all subjects. “ Was the grapheme-phoneme correspondence checked for all children and for all combinations present in the PW?
Response: We checked if the children immediately recognized all individual letters and did not confound letters e. g. p and q, b and d, m and n. We could not check all three letter combinations, as there are several thousands of them.
Reviewer 2:“Poor readers often fixate a word to be read longer than good readers do because they are aware that they need a longer fixation time.” How are they aware? The logic seems unclear here.
Response: I added the passage „…because they notice that they have not yet recognized the word to be read…”. (Discussion, p. 20).
Reviewer 2:“As word recognition improves with prolongation of fixation time, improvement of word recognition requires prolongation instead of shortening [32, 77-79] of the fixation time”: again, it seems that there is a problem in the writing here. May be “Results support the hypothesis that improvement of word recognition requires prolongation instead of shortening [32, 77-79] of the fixation time.”
Response: Has been reworded as requested (p. 20).
Reviewer 2:“The least letters were misread at the beginning of the word regardless of word length.”: could be simpler to write “children made less reading errors for letters at the beginning than at the end of the words.” Except for 6-letters words: pls comment.
Response 1: Has been reworded as requested (p. 21).
Response 2: For 6-letter pseudowords the difference between the rate of misread letters was always p>0.1. I can only speculate that this is just a statistical effect because readers who were tested with 6-letter pseudowords were not extremely bad readers. They were already able to read 5-letter pseudowords and did not misread many 6-letter pseudowords. Therefore, the comparison presumably did not reach a p-value smaller than 0.01. I am very reluctant to interpret p-values, as it has received a lot of criticism in modern statistics.
Reviewer 2:“Therefore, reduced ability to simultaneously recognize all letters in a pseudoword should not be regarded as a consequence of an impaired visual attention span [61, 62, 65, 66, 80-83]. Many poor readers improved their ability to simultaneously recognize a string of letters by applying a longer fixation time, i.e., an increased temporal summation [73-76].” This argument is unclear because a longer fixation time may induce more efficient visual attention processes: pls comment.
Response: I have added the text that highlights this aspect on p. 20: The finding that readers improve when they extend their fixation times shows that attention does not decrease during fixation and that the readers can maintain their attention for the required fixation time. This contradicts the assumption that poor reading is due to an attention deficit.
Reviewer 2:“Some children were unable to recognize pseudowords consisting of 4 or 5 letters even if the fixation time was prolonged up to 500 ms. Pls specify how many children?
Response: The number of children (19) has been added.
Reviewer 2: p.10, bottom: “Such a reading strategy must take into account the individual’s abilities to simultaneously recognize a given number of letters, the fixation time and speech-onset time needed to recognize and pronounce a given word segment. “ As mentioned in the general comment above, this is a very good point.
Response: Thank you
Reviewer 2: Figure 2: it seems that there are some mistakes in the legend: Figure 2A: “The speech spectrogram (E) shows that the subject pronounces more slowly during computer-guided reading.” The speech spectrograms seem to be C and F (rather than E) with F slower than C. Similar comment for Figure 2B: “
Response: The mistakes have been corrected (p.28).
Reviewer 2: As mentioned in the general comment a general conclusion is needed with a summary of the main findings, their interpretation and the perspectives for future work.
Response: I have added a conclusion and perspectives for future research.
Round 2
Reviewer 1 Report
Response to reviewer 1
Reviewer 1: The introduction starts with unclear and fragmented criteria for dyslexia. I recommend authors either to use the full-cited text or translate the criteria to a full sentence.
Response: The text has been reworded.
Reviewer 1: Furthermore, the theoretical underpinnings for this research are thin.
Response: The history of science shows that important scientific questions may be asked and answered without theoretical underpinnings. The question „why does the apple fall from the tree“ or „what is the nature of light“ can be asked without any knowledge of a physical theory. Nevertheless, the resulting research may lead to the development of Newtonian mechanics or quantum mechanics. Furthermore, the referee´s criticism is vague because it is not clear what the referee understands by „theoretical underpinnings“. If one specifies the concepts „theory“ or „theoretical“, the criticism proves to be unjustified.
One type of scientific theories may consist of assertions about the applicability of mathematical structures, such as Einstein's field equations on objects in space and time or Schroedinger's wave equation on photons and electrons. After the applicability of these theories had been proven many times, relativity theory became relativity mechanics and quantum theory became quantum mechanics. Then, these theories were no longer theoretical.
Statements that result from experimental findings but have not yet been tested experimentally, may also be termed „theories“. In dyslexia research, these are statements such as "the distribution of visual acuity in the retina, temporal summation, simultaneous recognition, the field of attention, eye movements etc., may have a fundamental role in reading." The objectives and methods employed in the present study are based on a number of experimental results as evinced by more than 100 references. The references also show that the methods used in the present study rest on findings about the distribution of visual acuity in the visual field and its physiological and psychophysical basis, eye movements, experimental results about the field of attention, simultaneous recognition, temporal summation of visual stimuli, and on the successful use of these methods in earlier studies. The questions investigated in the present work are therefore very well founded by experimental findings and „theories“ in the latter sense of the word.
Response to the author: Although, I sincerely grand every author a scientific breakthrough, and am willing to believe that every author has written his/her masterpiece, nevertheless there are some academic ground rules. Science indeed starts with curiosity, and asking questions. However, only in cases where one is the first to ask such questions one can claim that there is no theory yet. In other cases bringing science further requires that we clearly formulate expectations (hypothesis) based upon a set of reasoned premises, which may (or may not) already have developed into a theory. To bring science a step further these assumptions (or theory) needs to be tested, so that they can be discussed. In order to do so, it is important to be clear on the theoretical notions behind the hypotheses. Therefore, in every paper, transparency is needed about what, why and how things took place in the study and how findings can be explained based on theoretical assumptions. We do not just speculate on individual notions, we bring them to the ‘scientific forum’ in the form of a publication and treat criticism with respect.
Reviewer 1: A clear research question is missing in the introduction. It’s stays unclear why two experiments has been set up, what the aim of these experiments are and which hypotheses are tested.
Response to the author: It is common practice that an author indicates how the text has been reworded and that the response is a concrete answer to the comment. In this revised version questions are added that give an answer to the aim: “The present study investigated whether reading capacity can be improved in one single session if (1) only the reading strategy is changed, (2) no training is conducted that may improve the ability to expand the field of attention, or to focus attention, and (3) if no eye movement training is carried out.”. However, hypotheses are still lacking.
Reviewer 1: In this section 2.1.1 the participants, the methods and materials merge into each other. Make a clear distinction between these aspects.
Response: A new caption has been added.
Reviewer 1: With respect to the participants, there is a huge difference in age and graders. On what theoretical ground is chosen for such a great difference in age and graders. In the last sentence of the first paragraph of section 2.1.1. Authors claim that ‘the reading disabilities were not based on lack of teaching or inadequate educational instructions’. On what ground can they claim this? How are the children recruited, how many schools were involved?
Response: There is no reason to limit the children who participated to a given grade. If only children in a given grade participated one could argue that the results only apply to children in this grade. From a methodological perspective, it is essential that the children in the therapy group correspond to the children in the control group. This can only be achieved if the groups are paralleled i. e. if children assigned to the control group are selected according to their abilities tested in the pseudoword test. Assignment of children to the therapy group and the control group is described in detail in the methods section of Experiment 2 (page 14). I have added text that highlights this aspect on page 18.
As described on page 6, the children were recruited according to their performance in the Zuerich Reading Test if their reading abilities were 1.5 or 2 standard deviations below the performance of children in the same grades.
All children attended Bavarian primary schools and received the same reading instructions as the good readers.
Reviewer 1: The method is described in great detail, however a theoretical underpinning for is method is lacking.
Response: The criticism is unfounded. The methods employed in the present study are based on a number of experimental studies as evinced by more than 100 references. The references also show that the methods used in the present study rest on research about the distribution of visual acuity in the visual field, eye movements, the field of attention, simultaneous recognition, temporal summation of visual stimuli, and on the successful use of these methods in earlier studies.
Response to the author: It might be true that methods employed in the present study are based on a large number of experimental studies; however, we do not see these papers referenced as would be appropriate in the method section.
Reviewer 1: The statistics section lack body and content.
Response: What does it mean that the statistic section lacks body and content. The Wilcoxon test and the Bonferroni-Holm corrections are well known and widly employed. Their use does not require justification. The use of Cohen effect size statistics is the statistical method of choice to demonstrate the therapy effect and meets the requirements of the American Statistical Association. I have added the mathematics of this statistics on page 17.
Response to the author: The statistic section on page 4 only contains one sentence: ‘Rates of reading mistakes were compared using the Bonferroni-Holm corrected Wil-331 coxon-test.”. What I meant with ‘it lacks body and content’ is that one should describe not only which test is used, but what and how it has been done, and compared to what? The description of the use of Cohen effect size statistics is the statistical method of choice to demonstrate the therapy effect that meets the requirements of the American Statistical Association is still lacking. It should be clearly described and referred. Only adding the mathematics of this statistics in the reference list on page 17 (page 17 in my version) is insufficient. How should the reader be able to link this literature to this section?
Reviewer 1: Authors claim that the results suggest that poor reading is caused by an inappropriate eye movement strategy and a reduced ability to simultaneously recognize a sequence of letters. This is a huge claim based on a research without strong theoretical underpinnings and executed on a very small and a diverse research group of dyslectic children and a control group of non-dyslectic children is lacking.
Response: That poor reading is caused by an inappropriate eye movement strategy and a reduced ability to simultaneously recognize a sequence of letters is definitely not „a huge claim based on a research without strong theoretical underpinnings…“ This claim rests on the fact that visual acuity is only sufficiently high in the fovea and a small paravoveal area and drops dramatically towards the periphery. Therefore, eye movements must shift the words to be read in the center of the visual field where acuity is sufficient. I have added a text that highlights this aspect (pp. 22-23).
The literature concerning the field of attention and simultaneous recognition cited in the text also shows that the area in which we can recognize letters is limited by the expansion of the field of attention. The underpinning of this finding rests on numerous experimental results, many of which are cited in the text. I have added text that highlights this line of research (pp.20-21). Balint (1909) already showed that subjects can narrow or widen their field of attention depending on the dimensions of the object they are watching. Poppelreuter (1917) has shown that patients with a normally extended visual field may be unable to recognize objects next to the object on which attention is focussed. Objects further off the fixation point were only recognized when fixation times were longer. Williams and Gassel (1962) showed that the visual field narrows if a subject directs his/her attention vigorously to a point in the middle of the perimeter used to assess the extension of the visual field. Many investigations (e. g. Chaikin 1962; Engel 1971; Ikeda and Takeuchi 1975) and numerous more recent articles have shown that the area of the retina in which many stimuli can be detected simultaneously narrows when the subjects must focus their attention in the middle oft he visual field. The area where items are detected widens if the subjects don´t need to focus their attention on a given point of the retina, and if they can can spread their attention in a wider area. These studies show that the number of items which can be detected or recognized simultaneously depend on how much attention must be focussed on the center of the visual field. Many investigations (e.g. Chaikin 1962; Engel 1971; Ikeda and Takeuchi 1975) and numerous more recent studies have shown that the retinal area in which many stimuli can be detected simultaneously narrows when the subjects must focus their attention on the middle of the visual field. Posner (1980) and Jonides 1981) showed that the field of attention can be shifted in any area of the visual field without exerting eye movements. We have shown in earlier studies (Werth 2006, 2018.2019; Klische 2007; and the present study) that dyslexics have different abilities to recognize a given number of letters in pseudowords. Our studies studies confirm that the ability to recognize a given number of letters improves when the presentation time is prolonged.
Regarding the repeated criticism of a lack of theoretical underpinnings, what was said above about theories applies. This criticism is baseless.
The criticism that the claim is based „… on a very small and diverse research group of dyslexic children…“ does not apply. The claim rests on the result of 5 independent studies (including the present one) in which 356 dyslexic children participated. Compared to other studies on dyslexia this is not a small, but one of the largest groups of children which has been investigated. The result was also repeatable in 5 independent studies. The studies are well founded and meet the requirements of the American Statistical Association according to which studies must have repeatable results. In an earlier eye movement study I have demonstrated that reading performance in children with dyslexia improves (effect size Hedges g = 1.4) more than in other therapies if the amplitudes of saccades are adjusted to the children´s abilities to simultaneously recognize a string of letters and the children execute more saccades of smaller amplitudes (Werth 2019).
Response to the author: In line with my previous comment, it is good to read that there is indeed theory for this statement and that is made clear to every reader.
The criticism that „… a control group of non-dyslectic children is lacking“ is incorrect from a methodological point of view. To test a therapy effect, dyslectics with and without therapy must be compared, not dyslectics with good readers. A study with good readers is an inappropriate control study for what was investigated in the present study. The questions adressed in the present study are (1) how many letters dyslexic children can recognize in a pseudoword, (2) whether their abilities to recognize pseudowords of a given length improve if fixation times are prolonged, and (3) which role do eye movements play in reading. The methods appropriate for answering these questions were described in the text. To investigate pseudowords of which length good readers can recognize does not answer questions adressed in the present paper.
Response to the author: The author takes offence at my comments while the comments were based on a version of a paper in which the goal of the study or research questions were lacking in the first place. In this revised version the author has added research questions on page 2: “The present study investigated whether reading capacity can be improved in one single session if (1) only the reading strategy is changed, (2) no training is conducted that may improve the ability to expand the field of attention, or to focus attention, and (3) if no eye movement training is carried out.”. However, the questions formulated in this response is not the same as formulated in this revised version. Thus to date, it stays unclear what is conducted and why in this study.
Author Response
Response to the author: The author takes offence at my comments while the comments were based on a version of a paper in which the goal of the study or research questions were lacking in the first place. In this revised version the author has added research questions on page 2: “The present study investigated whether reading capacity can be improved in one single session if (1) only the reading strategy is changed, (2) no training is conducted that may improve the ability to expand the field of attention, or to focus attention, and (3) if no eye movement training is carried out.”. However, the questions formulated in this response is not the same as formulated in this revised version. Thus to date, it stays unclear what is conducted and why in this study.
Response to reviewer: Thanks for the suggestion. I have reworded this passage on p. 2 hoping that it is more clear now.